# Predicting Remission among Perinatal Women with Depression in Rural Pakistan: A Prognostic Model for Task-Shared Interventions in Primary Care Settings

**DOI:** 10.3390/jpm12071046

**Published:** 2022-06-27

**Authors:** Ahmed Waqas, Siham Sikander, Abid Malik, Najia Atif, Eirini Karyotaki, Atif Rahman

**Affiliations:** 1Department of Primary Care & Mental Health, Institute of Population Health, University of Liverpool, Liverpool L69 7ZA, UK; siham.sikander@liverpool.ac.uk (S.S.); atif.rahman@liverpool.ac.uk (A.R.); 2Global Institute of Human Development, Shifa Tameer-e-Millat University, Rawalpindi 46000, Pakistan; 3Department of Public Mental Health, Health Services Academy, Chak Shahzad, Islamabad 44000, Pakistan; abid.malik@hdrfoundation.org; 4Rawalpindi Medical University, Rawalpindi 46000, Pakistan; 5Human Development Research Foundation, Islamabad, Pakistan; najia.atif@hdrfoundation.org; 6Department of Clinical Neuro- and Developmental Psychology, Vrije Universiteit Amsterdam, 1081 BT Amsterdam, The Netherlands; e.karyotaki@vu.nl

**Keywords:** perinatal depression, prognosis, prognostic modeling, nomogram, Pakistan

## Abstract

Perinatal depression is highly prevalent in low- and middle-income countries (LMICs) and is associated with adverse maternal and child health consequences. Task-shared psychological and psychosocial interventions for perinatal depression have demonstrated clinical and cost-effectiveness when delivered on a large scale. However, task-sharing approaches, especially in LMICs, require an effective mechanism, whereby clients who are not likely to benefit from such interventions are identified from the outset so that they can benefit from higher intensity treatments. Such a stratified approach can ensure that limited resources are utilized appropriately and effectively. The use of standardized and easy-to-implement algorithmic devices (e.g., nomograms) could help with such targeted dissemination of interventions. The present investigation posits a prognostic model and a nomogram to predict the prognosis of perinatal depression among women in rural Pakistan. The nomogram was developed to deliver stratified model of care in primary care settings by identifying those women who respond well to a non-specialist delivered intervention and those requiring specialist care. This secondary analysis utilized data from 903 pregnant women with depression who participated in a cluster randomized, controlled trial that tested the effectiveness of the Thinking Healthy Program in rural Rawalpindi, Pakistan. The participants were recruited from 40 union councils in two sub-districts of Rawalpindi and randomly assigned to intervention and enhanced usual care. Sixteen sessions of the THP intervention were delivered by trained community health workers to women with depression over pregnancy and the postnatal period. A trained assessment team used the Structured Clinical Interview for DSM-IV current major depressive episode module to diagnose major depressive disorder at baseline and post-intervention. The intervention received by the participants emerged as the most significant predictor in the prognostic model. Among clinical factors, baseline severity of core-emotional symptoms emerged as an essential predictor, followed by atypical symptoms and insomnia. Higher severity of these symptoms was associated with a poorer prognosis. Other important predictors of a favorable prognosis included support from one’s mother or mother-in-law, financial empowerment, higher socioeconomic class, and living in a joint family system. This prognostic model yielded acceptable discrimination (c-statistic = 0.75) and calibration to aid in personalized delivery of the intervention.

## 1. Background

The perinatal period is a transitional period to motherhood and a time when women are vulnerable to mental health problems, including depression [1]. Perinatal depression (PND) occurs during pregnancy or within the first year following delivery [2]. A systematic review and meta-analysis on PND (*n* = 37,294 mothers without prior history of depression) found a worldwide prevalence of 17% and an incidence of 12% [3]. Poor socioeconomic conditions worsen maternal mental health and potentiate its adverse effect on their children [4]. The association with socioeconomic adversity might also partly explain the high burden of PND in low- and middle-income countries (LMICs). In Pakistan, for instance, the prevalence of PND is estimated at 30–37% [5,6]. 

This high burden of PND has profound effects on women and their families [7]. It is associated with wide-ranging adverse outcomes [7], including a negative impact on mother–child relationship and infant cognitive, socioemotional, and physical development [7,8,9,10]. Perinatal depression has, therefore, been recognized as a global public mental health priority by the population health stakeholders, advocating the need for timely treatments [11]. However, most low- and middle-income countries do not possess the much-needed human resource and infrastructure for ensuring treatment for all the women with PND. Besides the lack of physical infrastructure, the gap in knowledge of healthcare professionals regarding PND is immense [12]. These hurdles combined with stigma for mental illnesses translates to a treatment gap of 90% in low- and middle-income countries (LMICs) [13].

The treatment of PND also poses several challenges specific to the perinatal period [14]. Pharmacological treatment for PND is discouraged by perinatal women mainly due to concerns of teratogenicity [15]. Furthermore, there is a dearth of clinical trials on the effectiveness of antidepressants in the perinatal period due to ethical concerns [15]. The limited clinical trial evidence for antidepressants available has shown only small improvements in perinatal depressive symptoms, sometimes accompanied by side effects and poor adherence [15]. Fortunately, research on psychosocial interventions to treat PND and other common mental disorders has gained momentum in recent years [13], leading to innovations in screening, prevention, and treatment [13]. Interventional research in this area has identified cognitive behavioral therapy (CBT) as an effective treatment for PND [16]. In regions where the scale-up of the CBT interventions is limited due to a lack of specialist facilities, funds, and shortage of human resources, task-sharing strategies can be utilized [13,17]. These task-sharing strategies employing non-specialist primary care workers and lay peers to deliver psychosocial interventions were found to be cost-effective in several countries, including Pakistan [13,18,19]. 

Despite their benefits, the delivery of task-shared interventions poses challenges inherent to the use of non-specialists for healthcare delivery. Task-sharing, by its nature, demands a narrower set of skills to deal with specific health issues. Depression is a heterogeneous condition, and challenges exist, even in the specialist domain, owing to the highly variable trajectory of the depressive symptoms and variable response to treatments [20,21]. Previous literature has shown that different patient subgroups do not respond similarly to specific treatments or similar interventions delivered in different formats [22,23]. This necessitates the stratification of patient groups that would benefit from a low-intensity intervention, such as counseling by a non-specialist or self-help, and those requiring a higher intensity psychotherapy (such as standard cognitive behavioral therapy) or pharmacotherapy. 

On the other hand, it would be essential to reap the benefits of early and effective intervention, as untreated depression increases the risk of further relapse, with more severe symptoms and poorer functioning as well as adverse development outcomes in the infant [24,25]. Conventionally, such decision making requires expert clinical knowledge and skills. Recent research has shown that this stratification may be achieved by using clinical decision-support systems based on traditional statistical or more advanced machine learning models [20,21,22]. As the LMICs seek to implement task-sharing widely, more research is needed to allow targeted dissemination of these interventions, thereby maximizing the benefit.

Much of the research on the development of prognostic models and clinical decision support systems is limited to the context of major depressive disorders in high-income countries [26]. There is a paucity of such models for PND in LMICs, especially in Pakistan, where scale-up for task-shared interventions is underway [27]. These clinical decision support tools can provide a resource to make informed choices for selecting candidates suitable for primary care mental health interventions and ensure a more significant impact of such interventions. This investigation aims to develop and validate an easy-to-implement clinical prediction tool to assess prognosis and treatment response in task-shared intervention programs in primary care settings. 

## 2. Methods

### 2.1. Study Design

The study design conforms to the Transparent Reporting of a multivariable prediction model for Individual Prognosis or Diagnosis (TRIPOD) guidelines [28]. We developed this prognostic model using data from a large-scale cluster randomized, controlled trial (cRCT) of the THP program delivered by community health workers in rural subdistricts of Gujar Khan and Kallar Syedan in Rawalpindi, Pakistan. Details of the study design have been presented in our previous publications [29,30]. The primary study was conducted in accordance with the Declaration of Helsinki and approved by the Institutional Review Board of The University of Manchester, UK. However, ethical review and approval for present investigation were waived due to the secondary nature of analysis. Written informed consent was received from all study participants. 

Briefly, the study area comprised two rural sub-districts, and within these 40 Union Councils (UC). These UCs form the smallest administrative units within the sub-districts, and were chosen as the unit of randomization in the cRCT. Each UC has a population of around 22,000 to 25,000. All community health workers from the primary care health centers catering to the UCs’ health needs were employed as delivery agents for the Thinking Healthy Program (THP) intervention. For inclusion in this cRCT, all married women aged 16 to 45 years residing in the UCs, and within their third trimester of pregnancy, were invited from April 2005 to March 2006. Exclusion criteria included severe medical and pregnancy-related illnesses requiring inpatient hospitalization, profound learning or physical disability, and psychosis. 

For assessment of PND, all eligible women underwent detailed clinical assessments by experienced psychiatrists, blind to the allocation status of the participants, using the cross-culturally validated structured clinical interview schedule (SCID) based on the Diagnostic and Statistical Manual of Mental Disorders (DSM-IV) [30]. Pregnant women diagnosed with PND were recruited into the trial, yielding a total sample size of 903 pregnant women across 40 UCs. The UCs were randomized to either receive THP or enhanced usual care. Those in the intervention group were delivered a session of THP intervention every week for four weeks in the last month of pregnancy, three sessions in the first postnatal month, and nine once-a-month sessions thereafter [30]. Mothers in the control arm received an equal number of visits with similar frequency by the community health workers. The content of these visits covered antenatal and postnatal preventive and promotive health care. Remission from depression was assessed at six months postnatal using the current depressive episode module of SCID administered by trained psychiatrists blind to the allocation status of the participants.

### 2.2. Predictor Selection

A battery of questionnaires was performed by a blinded assessment team at the baseline, providing a rich source of predictors for modeling. The data were divided into four broad categories: demographic characteristics, family structure and social support, socioeconomic status, and mental health indicators. Details on these variables is provided as Table 1. Predictors of interests included characteristics of mothers, such as age, education levels, household income, and physical health. Social support levels were assessed using the Multidimensional Scale of Perceived Social Support [31]. Socioeconomic status was assessed using employment and household income questions, a detailed asset questionnaire, and subjective assessments by the community health workers. Finally, assessments of mental health and functional impairment included the Hamilton Depression Rating Scale (HDRS) [32], Brief Disability Questionnaire (BDQ) [33], and the Global Assessment of Functioning scores (GAF) [34].

We also included the allocation of trial participants to either the THP intervention or enhanced care as usual group as a variable. Briefly, the THP [29,30] is a task-shared psychosocial intervention underpinned by cognitive-behavioral approaches, including cognitive restructuring, behavioral activation, and problem solving. It is an evidence-based manualized intervention endorsed by the World Health Organization as a low-intensity treatment for perinatal depression. The THP also addresses stigma toward perinatal depression and helps women with perinatal depression to identify and elicit social support networks. It comprises a total of 16 sessions including one or two introductory sessions, one weekly session for four weeks in the last month of pregnancy, and then nine monthly sessions during the postnatal period. Enhanced care as usual comprised psychoeducation of trial participants along with usual house visits conducted by the community health workers [30].

The choice of the predictors included in the prognostic model building process was based on our literature review of previous key papers exploring prognostic models of depression [20,21,23,26,35,36]. This strategy was further augmented by consensus opinion by clinical experts in the team. We included predictors that were easily assessed in the primary healthcare settings in rural Pakistan. In a similar exercise, Moriarty et al.’s review of prognostic models for major depressive disorder relapse and recovery revealed three critical domains of predictors [26]. Across these key papers, similar domains of variables predicting prognosis for depression emerged. For instance, disease-related variables, such as previous depressive episodes, presence of residual symptoms, higher baseline severity, and duration of index episode, were associated with a worse prognosis. Demographic factors, such as older age and living alone, and psychosocial predictors, such as exposure to stressful life events, disability, poor social support network, and interpersonal difficulties, also predicted poorer prognosis. Finally, biochemical tests, such as higher serum levels of the corticotrophin-releasing hormones and higher scores on symptom checklists for depression, were also associated with a poorer prognosis requiring intensive treatment strategies [20,21,23,26,35,36]. 

In addition to the review by Moriarty et al., we also considered results from two individual participant data meta-analyses (IPDMA) [22,37] conducted by Karyotaki et al. These IDPMAs utilized data from 11 RCTs of task-shared psychotherapies for PND. These analyses revealed that improvement in perinatal depressive symptoms depended on the severity of individual symptoms at baseline, especially psychomotor symptoms, tiredness, and sleep problems [22,37].

### 2.3. Model Building Strategy

All analyses were conducted in Stata version 17 (College Station, TX, USA). Logistic regression with cluster robust standard errors was utilized to build the model, where the outcome was a dichotomous variable of diagnosis of depression established using the SCID module post-intervention [38]. For predictors, the variables were defined as groups/blocks based on the nature of the constructs they measured. We did not use forward or backward selection methods. The blocks of variables included maternal characteristics, family structure and social support, socioeconomic status, and mental health assessments, including scores on GAF, BDQ, and HDRS. When two or more predictors assessing similar constructs were available, the choice of inclusion was based on the values of BIC and AIC, i.e., the constructs leading to a better model fit were retained. 

Using these criteria, decisions were made to choose between either total score on HDRS at baseline or its symptom dimensions, scores on MSPSS scale or variables such as family and support structure, and BDQ and GAF. Variables with regression coefficients close to 0 (<0.5) contributed little to the overall model and were dropped. We also assessed whether cubic spline transformations for continuous variables improved the model fit. Model adequacy and apparent validation were assessed using several fit statistics, including the Cox and Snell R^2^, AIC and BIC values, and the Brier score (adequate at <0.25) [39].

In contrast to previous modeling strategies, especially by Chondros et al. [20] and Karyotaki et al. [22,37], we did not use individual symptoms of PND in the model. We instead favored scores on symptom dimensions obtained after dimension reduction techniques to avoid multicollinearity in model building. Detailed analyses on the development of these symptom dimensions have been reported elsewhere [29]. Briefly, by applying dimension reduction and cluster analyses on individual items of the clinician rated HDRS, we elucidated four symptom dimensions for PND: Core emotional symptoms: depressed mood, anhedonia, loss of appetite, psychic anxiety, and somatic anxiety.Somatic symptoms: loss of weight, psychomotor retardation, hypochondriasis, suicidal ideation, and somatic symptoms.Insomnia symptoms: early, middle, and late insomnia.Atypical symptoms: hypersomnia, hyperphagia, and weight gain.

Model performance was assessed using the concordance c-statistics with bootstraps to calculate the 95% confidence intervals. A c-statistic greater than 0.65 was considered adequate [39]. A calibration plot was visualized to plot the observed depression diagnosis (y-axis) with predictions (x-axis) for ten high-risk groups. Perfect predictions lie on the line of identity. We also assessed calibration-in-the-large, ratio of expected and observed outcomes, and the value for calibration slope [39]. 

The relative importance of the predictors in the final logistic regression model was estimated using the dominance analysis approach. The relative important of each predictor is based on its contribution to the overall model fit statistic. It is an ensemble method that determines predictor importance by running multiple models containing each possible combination of predictors and aggregating results [40,41].

### 2.4. Internal Validity

To adjust for over-optimism, we applied the heuristic shrinkage factor calculated using the formula X^2^-df/X^2^, thus accounting for the number of predictors in the model building process. Internal validation was performed to obtain an unbiased estimate of the model’s predictive performance [39]. Internal validation was done using the bootstrap method. The model was replicated in the bootstrapped sample using the same method as delineated above. Apparent performance was calculated for the bootstrapped sample and test performance in the original sample. These estimates were then used to calculate optimism in the model. A total of 1000 replications were performed to obtain average optimism, and original model estimates were adjusted accordingly. 

### 2.5. Utility of Prognostic Tool

For use in field and clinical settings, the prognostic model was visualized as a nomogram. The nomogram was developed using the nomolog package in Stata v.17 (College Station, TX, USA). Nomograms provide a convenient approach to calculating output probabilities for predictive models based on logistic regression [39]. The final output of the nomogram is the probability of event (remission of depressive symptoms) ranging from 0 to 0.95.

A stakeholder group (*n* = 8) comprising local psychiatrists, psychologists, a mental health system expert, and people with lived experience based on their professional and personal experiences determined the acceptability of the prognostic tool for use in rural Pakistan. The stakeholder group was also consulted to provide a cut-off value on the nomogram’s probability of event scale. This cut-off value could be used in clinical practice to label cases as standard (yielding optimum response to THP) and complex (not responding to THP and requiring specialist assessment).

### 2.6. Sample Size Calculation

Post hoc sample size estimation was done using *pmsampsize* module in Stata v.17. A minimum of 708 participants with 248 events were found to be adequate assuming an outcome prevalence of 0.35, an EPP of 16.52, expected c-statistic of 0.75, and 15 parameters [42]. 

## 3. Results

Characteristics of 903 study participants are presented in Table 1. Outcome data about response to treatment at six months were available for 818 women (90.56%), which were used to develop the logistic regression model. Post-intervention, significant differences in rates of remission from depressive disorder were found among the intervention recipients (*n* = 321, 76.8%) compared to their control counterparts (*n* = 189, 47.3%). There were no missing values for any of the predictor variables.

Table 2 shows the estimated beta coefficients and optimism adjusted betas for the final model. Out of the 19 predictor variables, only nine were retained. Introduction of cubic splines for continuous variables did not improve the goodness of fit (not shown here). The final model yielded lower AIC (954.54) and BIC (1001.61) values than the initial model, revealing better goodness of fit. Cox and Snell R^2^ and Crag and Uhler’s R^2^ values were 16.7% and 22.7%, respectively. Brier score was estimated at 0.19, which is significantly less than the maximum value of 0.25. Hosmer and Lemeshow X^2^ (*p* = 0.67) and Pearson X^2^ values were statistically non-significant.

Among the predictors, allocation to the intervention arm emerged as the strongest predictor (Figure 1), followed by the severity of core symptoms of depression, maternal empowerment, living with a grandmother, symptoms of insomnia, living in a joint family system, and socioeconomic class. Severity of atypical symptoms emerged as the least important variable. High severity of depressive symptoms, poor maternal empowerment, poorer socioeconomic class were associated with a poor prognosis. And living in a joint family system and with mother or mother-in-law were associated with a better prognosis. Appendix A presents the questionnaire detailing the variables in the final logistic regression model.

The overall model performance was adequate. The model yielded a c-statistic of 0.75 (95% CI: 0.71 to 0.78), indicating good discrimination (Figure 2). Several calibration measures were used to assess whether the model predictions’ accuracy matched the observed data. In the development data, the ratio of expected and observed number of events was 99.7%, CITL at 0, and a calibration slope of 1. Calibration plot with LOWESS smoother indicated a good-fitted model by visualizing the observed and expected probability of outcome among 10 high-risk groups (Figure 3). 

Internal validation was performed in the bootstrapped study sample. A heuristic shrinkage factor of 0.93 was applied to adjust beta coefficients for overoptimism (Table 2). In the bootstrapped sample, the optimism adjusted indices revealed comparable discrimination and calibration: optimism adjusted c-statistic (0.73), optimism adjusted CITL (0.0009), and optimism adjusted C-slope (0.94).

The stakeholders considered the prognostic tool and the resulting nomogram (Figure 4) to be acceptable for use in community and clinical settings in Pakistan. A consensus was reached to use the cut-off value of 0.40 on the nomogram’s probability of event scale. This value was chosen because it corresponded to participants’ reporting worse symptom scores on HDRS, structure of social network (living with mother or mother-in-law, structure of household), and socioeconomic variables (economic status and empowerment). Respondents scoring 0.40 or below could be considered as complex cases and referred to specialist care rather than be offered THP. This strategy was also encouraged by mental health systems experts to avoid straining the finite specialist healthcare resources in Pakistan. Furthermore, future investigations considering variables pertaining to stressful and traumatic life events and biochemical events were encouraged. 

## 4. Discussion

The present study presents a prognostic model for predicting remission from PND among pregnant women in rural Pakistan. The use of the presented logistic regression-based nomogram is an efficient approach to predicting the prognosis of PND in Pakistan. It can also be used to select the best candidates for the THP for perinatal depression and channel clients less likely to respond to THP to specialist services. This is important, as the longer a depressive episode persists, the worse are the outcomes for the infant. 

The final logistic regression model comprised eight predictor variables, including treatment condition, socioeconomic class, family structure, and severity of heterogeneous symptom dimensions of depression. Women yielded a better prognosis for PND if they had received the THP intervention, lived in a joint family system and with infants’ grandmothers, belonged to a higher socioeconomic class, and reported lower severity scores on dimensions of core emotional, insomnia, and atypical symptoms. We show that this model has a good discriminatory ability and calibration. Due to the ease of recording this information, it can be deployed in primary care settings after undergoing external validation procedures.

The model’s overall performance was adequate with a pseudo R^2^ of 22.9%, which is comparable to previously developed prognostic models [39]. The R^2^ indicates the predictability of the outcome, and models that explain more than 20% of the variability have the potential to be clinically useful and warrant further evaluation and development [39]. It also yielded a c-statistic of 0.75, which is in the range of 0.60 to 0.85 for models predicting depression onset, treatment outcome, and relapse [20,21,23,43]. Our findings build upon previous research where the concept of heterogeneity in depression has been leveraged to predict treatment outcomes accurately [29]. Besides accounting for heterogeneity in the clinical presentation for PND, this model also incorporates short and easy-to-measure constructs of sources of social support, relationship status and empowerment, and socioeconomic class. These variables have been shown to affect the trajectory and hence the prognosis of PND and major depressive disorder [26,31,44]. It is, however, to be noted that the current modeling strategy included subjective assessments of socioeconomic class by the community health workers and sources of social support using categorical questions. Inclusion of these variables in the model yielded better performance than objectively measured social support levels using the MSPSS scale and self-reported income levels and status of employment. 

In the present prediction model, symptom dimensions of PND were found to be more important predictors than total scores on HDRS at baseline. This emphasizes the importance of heterogeneity in PND and major depressive disorder in general. In this context, previous research has shown that depressive disorders are highly heterogeneous, with varying clinical presentations [1,45,46]. Our previous work [29] noted that for PND, the symptoms of HDRS clustered together into 365 different combinations. Research on heterogeneity in PND and its link with maternal morbidity, child outcomes, and treatment considerations are lacking. Nonetheless, preliminary research has shown that high burden symptom trajectories of PND are associated with poor child outcomes [44]. For treatment considerations, research on major depressive disorder has shown variable response of antidepressant treatment to different symptom profiles [21,23,47]. Recent studies have shown that item-level predictions for treatment response may outperform sum scores on depression rating scales [48], with important predictors of task-shared treatment response being psychomotor symptoms, insomnia, and fatigue [22]. However, it must be noted that some of the symptoms of depression reported on the HDRS scale such as fatigue may be associated with physical health during pregnancy and postpartum itself.

The data for the current analyses were curated from a high-quality cluster randomized controlled trial comparing a task-shared CBT-based intervention with enhanced care as usual in rural settings in Pakistan. The Thinking Healthy Program is a multicomponent CBT based intervention delivered by lady health workers and peers [13,18,19]. This intervention is currently being scaled up in Pakistan as part of the President’s program to promote the mental health of Pakistanis [27]. This program ensures the delivery of the THP program to women at high risk of PND in Pakistan [27]. Another important aspect of this plan is to digitize the delivery of the THP intervention to circumvent the shortage of mental health specialists in Pakistan. Developments are currently underway to develop and test the delivery of the THP using mental health apps by primary care workers and peers [27]. The presented prognostic model has the potential to augment the clinical utility of this program by matching this therapy to correct candidates. 

## 5. Implications for Future Practice

The present analyses present a novel prognostic model and an easy-to-use nomogram suitable for assessing the prognosis of PND in rural Pakistan. It leverages the concept of heterogeneity in the presentation of PND to yield the probability of remission in PND. It also utilizes care-as-usual approaches and low-intensity psychosocial approaches to stratify patients according to the treatment they would best respond to. In this way, patients requiring intensive treatment strategies could be referred to tertiary care centers or specialist mental health services at the outset, while those with favorable responses to low-intensity and cheaper treatments could be identified early, leading to a more efficient system of care.

Timely treatment and support are vital due to associated maternal and infant morbidity. We opine that this challenge could be mitigated using clinical decision support tools, especially by coupling them with electronic health applications. Several investigators have developed and validated such clinical decision-support systems based on either statistical modeling or machine learning approaches [20,21]. The utility of these decision support systems has been shown in pragmatic trials conducted in high-income countries [36,49]. These trials presented these tools’ clinical and cost effectiveness, where patient groups report better outcomes when stratified than enrolled in stepped-care approaches [36,49].

## 6. Strengths and Limitations

There are several strengths of this study. It utilizes data from a high-quality, pragmatic, cluster randomized, controlled trial that tested a psychosocial approach for PND in a real-world setting. Candidate predictors are patient-reported and do not contain any sensitive questions. The nomogram is also easy to use by the non-specialist workforce after minimal training. However, external validation and further randomized, controlled trials are needed to ascertain the effectiveness of using this model before large-scale implementation. Furthermore, we also encourage investigators to develop prognostic models using datasets with more treatment arms to suggest alternative treatment options. A nomogram accounting for more treatment strategies would prove to be a more robust tool for precision mental health delivery. The present study is well-designed with access to a range of clinical and psychosocial variables. However, we encourage future investigators to account for more variables such as life events and biochemical indicators such as cortisol and dysregulation of the HPA-axis.

## Figures and Tables

**Figure 1 jpm-12-01046-f001:**
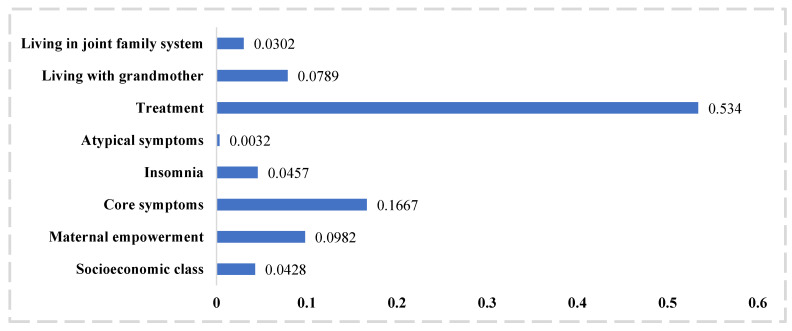
Importance of predictors in the final model presented as standardized dominance statistic.

**Figure 2 jpm-12-01046-f002:**
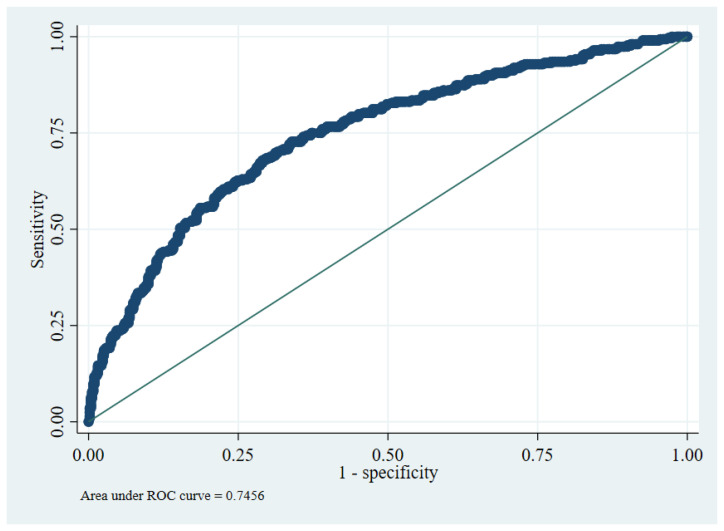
ROC curve presenting discriminatory ability of the prognostic model.

**Figure 3 jpm-12-01046-f003:**
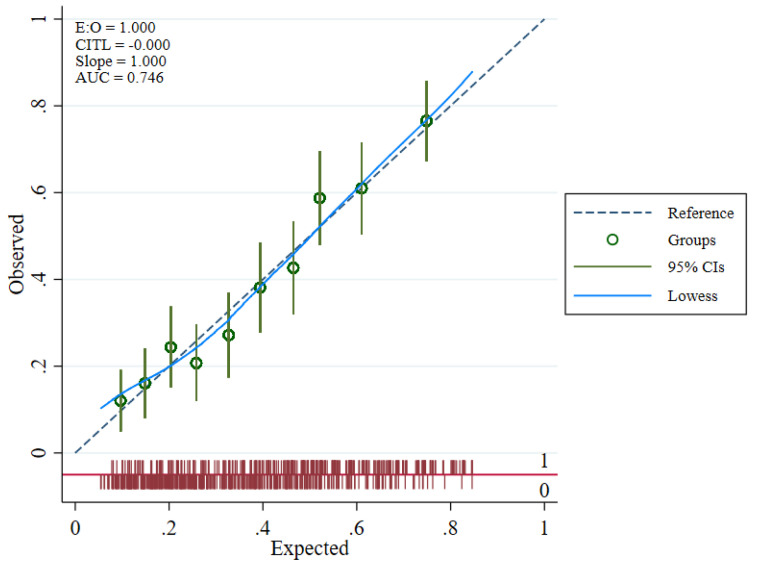
PMCA plot presenting calibration of the prognostic model by comparing observed and expected outcomes.

**Figure 4 jpm-12-01046-f004:**
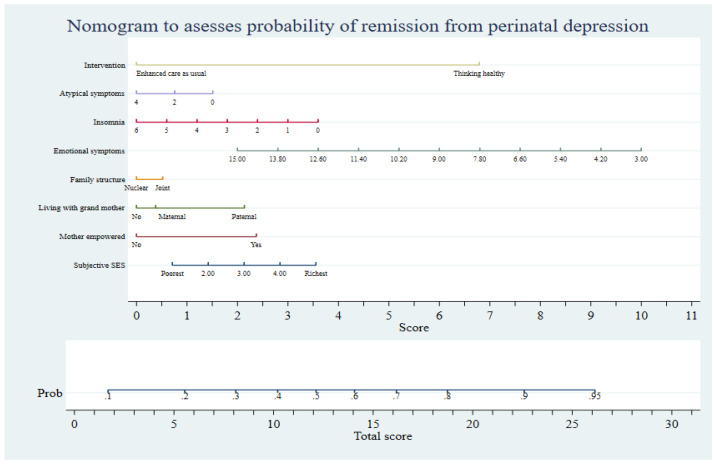
Nomogram presenting the prognostic model for use in the primary care setting.

**Table 1 jpm-12-01046-t001:** Description of candidate predictors for inclusion in the initial prediction model.

Characteristics	Subgroup	Mean (SD)	Frequency	Percentage
Outcome				
Perinatal women with depression post-intervention assessed using the DSM-IV criteria	Enhanced Usual Care		211	52.8%
	Thinking Healthy Program		97	23.2%
Maternal demographic characteristics				
Mother age at baseline		26.74 (5.11)		
Maternal education level		4.06 (4.011)		
Paternal education level		7.02 (3.965)		
Socioeconomic condition				
Socioeconomic class	Richest		12	1.3%
	Rich		81	9.0%
	Normal		343	38.0%
	Poor		270	29.9%
	Poorest		197	21.8%
Household debt	No		371	41.1%
	Yes		529	58.6%
	Not reported		3	0.3%
Sufficient money for food	No		120	13.3%
	Yes		783	86.7%
Sufficient money for basic needs	No		189	20.9%
	Yes		714	79.1%
Financial Empowerment	Not empowered		425	47.1%
	Empowered		478	52.9%
Family structure				
Parity	0		171	18.9%
	1 to 3		520	57.6%
	More than 4		212	23.5%
Family structure	Nuclear		373	41.3%
	Joint		530	58.7%
Living with mother or mother-in-law	No		451	49.9%
	Maternal		59	6.5%
	Paternal		393	43.5%
Perceived levels of social support		45.04 (16.44)		
Clinical profile				
Hamilton depression scores at baseline		14.63 (4.09)		
Chronicity (months)		5.15 (9.08)		
Disability scores (BDQ)		8.21 (2.69)		
Global assessment of functioning (GAF)		62.05 (5.22)		
Insomnia symptom dimension of HDRS		2.33 (1.81)		
Somatic symptom dimension of HDRS		2.47 (1.47)		
Core emotional symptoms dimension of HDRS		8.37 (.65)		
Atypical symptoms dimension of HDRS		0.17 (0.57)		
Major perinatal life events				
Child death	None		518	57.4%
	Yes		385	42.6%
Still birth	None		607	67.2%
	Yes		296	32.8%
Treatment				
Enhanced Usual Care			440	48.7%
Thinking Healthy Program			463	51.3%

**Table 2 jpm-12-01046-t002:** Logistic regression analyses for predicting remission in depression.

Variables	Coefficients	Robust S.E.	Coefficients Adjusted for Optimism	z	Predictor Importance	*p*	95% CI
Socioeconomic class	0.1495407	0.0909545	0.139072851	1.64	6	0.1	(−0.0287269 to 0.3278083)
Maternal empowerment	−0.4992032	0.1969697	−0.464258976	−2.53	3	0.011	(−0.8852568 to −0.1131496)
Living with mother or mother-in-law							
Maternal	−0.0800688	0.3280223	−0.074463984	−0.24	4	0.807	(−0.7229807 to 0.5628432)
Paternal	−0.4495611	0.2246094	−0.418091823	−2		0.045	(−0.8899834 to −0.009388)
Family structure	−0.1090834	0.230452	0.101447562	−0.47	7	0.64	(−0.5607609 to 0.3425942)
Symptom dimensions of depression							
Core emotional symptoms	0.1402178	0.0332881	0.130402554	4.21	2	<0.001	(0.0749744 to 0.2054612)
Insomnia	0.1261017	0.0478054	0.117274581	2.64	5	0.008	(0.0324048 to 0.2197985)
Atypical symptoms	0.0794525	0.1272977	0.073890825	0.62	8	0.533	(−0.1700464 to 0.3289514)
Treatment	−1.4283	0.208072	−1.328319	−6.86	1	<0.001	(−1.836114 to −0.2638574)
Constant	−1.372766	0.5657802	−1.31	−2.43		0.015	(−2.481675 to −0.2638574)
Linear predictor = −0.61 (SD 0.98); Linear predictor adjusted for optimism = −0.599 (0.91)							

## Data Availability

The data associated with this manuscript are available upon request from the corresponding author.

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
