# Peer review of "Predicting Remission among Perinatal Women with Depression in Rural Pakistan: A Prognostic Model for Task-Shared Interventions in Primary Care Settings"

_jpm, 2022, doi:10.3390/jpm12071046_

Round 1
Reviewer 1 Report
This is a very interesting paper focused on an investigationg to predic remission in perinatal women suffering from depression. The authors tried to design a prognostic model for task-shared interventions in primary care settings.
The paper is well-written and of interest for the readers. However, I recommend the following minor changes.
In the abstract section the authors are opening the topic with a brief background about mental health training. I would also add a brief sentence on perinatal depression. The main aims of the project: design of a prognostic model for interventions in primary care, should beclarified in the abstract.
In the abstract the authors report that they used the DSM-4 criteria for depression. DSM-4 should be changed to DSM-IV. They mention that they used the module to diagnose depression. What diagnosis? Major depressive disorder? Episode of major depression? This should be clarified.
In the background section the authors provide a definition for perinatal depression, which occurs during pregnancy and within the first year postpartum. What does major and minor episodes mean? Could the authors provide an explanation for this?
Pharmacological treatment is not contraindicated in perinatal depression. I woudl prefer to clarify it by reviewing (in brief) the evidence for that. Shared decision treatment models seem to be applied in these populations. Evidence for "major" and "minor" depresion should be stratified.
In table 1, GAF has been used as an abbreviation for insomnia. This should be corrected and applied for the previous line (Global Assessment of Functioning).
Table 2 has been called "Finalized logistic regression". I would prefer to rename it as "Logistic regression analyses for predicting...".
Hamilton Depression Rating Scale, in the discussion section, has been used several times in the discussion section. I would use it once and then the abbreviation.
Author Response
Dear Editor,
We are very grateful to you and the reviewers for their excellent feedback on the manuscript. We have incorporated all the comments in the revised manuscript and provide line-by-line responses below. We believe this has substantially improved the quality of the manuscript.
We look forward to your final decision in due time.
Best wishes,
Dr. Ahmed Waqas
Corresponding author
Reviewer 1
Comment 1:
This is a very interesting paper focused on an investigating to predict remission in perinatal women suffering from depression. The authors tried to design a prognostic model for task-shared interventions in primary care settings.
The paper is well-written and of interest for the readers. However, I recommend the following minor changes.
Response
We are grateful to the reviewer for their kind words and feedback. We have incorporated all the suggestions in the manuscript as described below.
Comment 2:
In the abstract section the authors are opening the topic with a brief background about mental health training. I would also add a brief sentence on perinatal depression. The main aims of the project: design of a prognostic model for interventions in primary care, should be clarified in the abstract.
Response:
Thank you for this excellent comment. We have now revised our abstract section by adding a sentence on perinatal depression. It reads as,
“Perinatal depression is highly prevalent in low and middle-income countries (LMIC) and is as-sociated with adverse maternal and child health consequences. Task-shared psychological and psychosocial interventions for perinatal depression have demonstrated clinical and cost-effectiveness when delivered on a large scale in LMIC.”
We have further clarified the aims of the project. It reads as,
“Therefore, the present investigation posits a prognostic model and a nomogram to predict the prognosis of perinatal depression among women in rural Pakistan. Such a nomogram can be used to deliver stratified mental healthcare in communities by identifying those women who respond well to task-shared interventions and their counterparts requiring specialist care.”
Comment 3:
In the abstract the authors report that they used the DSM-4 criteria for depression. DSM-4 should be changed to DSM-IV. They mention that they used the module to diagnose depression. What diagnosis? Major depressive disorder? Episode of major depression? This should be clarified.
Response:
DSM-4 has been replaced with DSM-IV. We have clarified the use of SCID. It reads as, “A trained assessment team used the Structured Clinical Interview for the DSM-IV current major depressive episode module to diagnose major depressive disorder at the baseline and post-intervention.”
Comment 4:
In the background section the authors provide a definition for perinatal depression, which occurs during pregnancy and within the first year postpartum. What does major and minor episodes mean? Could the authors provide an explanation for this?
Response:
We have removed the terms major and minor depressive episodes. This is primarily for two reasons: a. explaining these would hinder the flow of the manuscript and b. the term minor depressive episode has been removed from the DSM-V. It was defined in DSM-IV TR as a mood disorder that does not meet the full criteria for major depressive disorder but at least two depressive symptoms are present for a long time.
Comment 5:
Pharmacological treatment is not contraindicated in perinatal depression. I would prefer to clarify it by reviewing (in brief) the evidence for that. Shared decision treatment models seem to be applied in these populations. Evidence for "major" and "minor" depression should be stratified.
Response:
This is a very thoughtful comment. We have “toned down” the language for use of pharmacological treatment for perinatal depression. However, we have also added evidence based on recent Cochrane systematic review on pharmacological treatment for postnatal depression [1].
We added the following passage:
“The treatment of PND also poses several challenges specific to the perinatal period [14]. Pharmacological treatment for PND is discouraged by perinatal women mainly due to concerns of teratogenicity [15]. Furthermore, there is a dearth of clinical trials on the effectiveness of antidepressants in the perinatal period due to ethical concerns [15]. And the little clinical trial evidence for antidepressants available has shown only small improvements in perinatal depressive symptoms sometimes accompanied by side effects and poor adherence [15].”
Reference:
- Brown, J. V. E.; Wilson, C. A.; Ayre, K.; South, E.; Molyneaux, E.; Trevillion, K.; Howard, L. M.; Khalifeh, H., Antidepressant treatment for postnatal depression. Cochrane Database of Systematic Reviews 2020.
Comment 6:
In table 1, GAF has been used as an abbreviation for insomnia. This should be corrected and applied for the previous line (Global Assessment of Functioning).
Table 2 has been called "Finalized logistic regression". I would prefer to rename it as "Logistic regression analyses for predicting...".
Response:
These corrections have now been made in the revised manuscript.
Comment 7:
Hamilton Depression Rating Scale, in the discussion section, has been used several times in the discussion section. I would use it once and then the abbreviation.
Response:
This has now been modified in the manuscript.
Reviewer 2 Report
Należy omówić The THP program and Enhanced Usual Care should be discussedprogram THP i Enhanced Usual Care.
The conclusions are not supported by the results.
Author Response
Dear Editor,
We are very grateful to you and the reviewers for their excellent feedback on the manuscript. We have incorporated all the comments in the revised manuscript and provide line-by-line responses below. We believe this has substantially improved the quality of the manuscript.
We look forward to your final decision in due time.
Best wishes,
Dr. Ahmed Waqas
Corresponding author
Reviewer 2
Comment 1:
The THP program and Enhanced Usual Care should be discussed.
Response
We have now added a detailed section in methods describing the THP program and Enhanced Usual Care.
It reads as,
“We also considered the variable on allocation of trial participants to either the Thinking Healthy Programme (THP) intervention or active enhanced care as usual group. Briefly, the THP [29, 30] is a task-shared psychosocial intervention underpinned by cognitive-behavioural approaches including cognitive restructuring, behavioural activation, and problem solving. It is an evidence based manualized intervention endorsed by the World Health Organization as a low-intensity treatment for perinatal depression. The THP also addresses stigma toward perinatal depression and helps women with perinatal depression to identify and elicit social support networks. It comprises a total of 16 sessions including one or two introductory sessions, one weekly session for four weeks in last month of pregnancy, and then monthly sessions for nine months of pregnancy. Enhanced care as usual comprised psychoeducation of trial participants along with usual house visits conducted by the community health workers [30].”
Comment 2:
The conclusions are not supported by the results.
Response
We think our conclusion of the tool being robust and valid for use in Pakistan is quite justified. However, we further strengthen our sections in conclusion pertaining to use of the tool in precision care.
For this, we held a stakeholder consensus meeting to comment on acceptability for use in clinical and community settings in Pakistan. We have added in following details the methods and results section:
Methods:
2.5. Utility of Prognostic Tool
For use in field and clinical settings, the prognostic model was visualized as a nomogram. The nomogram was developed using the nomolog package in Stata v.17. Nomograms provide a convenient approach to calculating output probabilities for predictive models based on logistic regression [39]. The final output of the nomogram is the probability of event (remission of depressive symptoms) ranging from 0 to 0.95.
A stakeholder group (n=8) comprising psychiatrists, psychologists, mental health system experts, social workers, and people with lived experience judged the acceptability of the prognostic tool for use in rural Pakistan. The stakeholder group was also consulted to provide a cut-off value on the nomogram’s probability of event scale. This cut-off value could be used in clinical practice to label cases as standard (yielding optimum response to THP) and complex (not responding to THP and requiring specialist assessment).
Results
“The stakeholders considered the prognostic tool and the resulting nomogram to be ac-ceptable for use in community and clinical settings in Pakistan. A consensus was reached to use the cut-off value of 0.40 on the nomogram’s probability of event scale. This value was chosen because it corresponded to participants’ reporting worse symptom scores on HDRS, structure of social network (living with grandmother, structure of household) and socioeconomic variables (empowerment). Respondents scoring 0.40 or below could be considered as complex cases and referred to specialist care rather than be offered THP. Furthermore, future investigations considering variables pertaining to stressful and trau-matic life events and biochemical events were encouraged.”